# Could Subtle Obstetrical Brachial Plexus Palsy Be Related to Unilateral B Glenoid Osteoarthritis?

**DOI:** 10.3390/jcm10061196

**Published:** 2021-03-12

**Authors:** Alexandre Lädermann, Hugo Bothorel, Philippe Collin, Bassem Elhassan, Luc Favard, Nazira Bernal, Patric Raiss, George S. Athwal

**Affiliations:** 1Division of Orthopaedics and Trauma Surgery, La Tour Hospital, 1217 Meyrin, Switzerland; 2Faculty of Medicine, University of Geneva, 1211 Geneva, Switzerland; 3Division of Orthopaedics and Trauma Surgery, Department of Surgery, Geneva University Hospitals, 1205 Geneva, Switzerland; 4Research Department, La Tour Hospital, 1217 Meyrin, Switzerland; hugo.bothorel@latour.ch; 5Centre Hospitalier Privé Saint-Grégoire (Vivalto Santé), 35768 Saint- Grégoire, France; docphcollin@gmail.com; 6Department of Orthopaedic Surgery, Mass General Hospital, Boston, MA 02114, USA; belhassan@partners.org; 7Université de Tours, Service d’Orthopédie, CHU Trousseau, 37000 Tours, France; luc.favard@univ-tours.fr; 8Department of Orthopedic Surgery, Clinica Alemana—Universidad del Desarrollo, 7650568 Santiago, Chile; nazirabbader@gmail.com; 9OCM (Orthopädische Chirurgie München), Steinerstrasse 6, 81369 Munich, Germany; patric.raiss@gmail.com; 10Roth McFarlane Hand and Upper Limb Center, St Joseph’s Health Care, London, ON N6A 4L6, Canada; gathwal@uwo.ca

**Keywords:** shoulder pathology, B glenoid osteoarthritis, inclination, neurological lesion, muscular disbalance, delivery, childbirth, OBPP

## Abstract

Background: Several factors associated with B glenoid are also linked with obstetrical brachial plexus palsy (OBPP). The purpose of this observational study was to determine the incidence of OBPP risk factors in type B patients. Methods: A cohort of 154 patients (68% men, 187 shoulders) aged 63 ± 17 years with type B glenoids completed a questionnaire comprising history of perinatal characteristics related to OBPP. A literature review was performed following the Preferred Reporting Items for Systematic Review and Meta-Analysis (PRISMA) to estimate the incidence of OBPP risk factors in the general population. Results: Twenty-seven patients (18%) reported one or more perinatal OBPP risk factors, including shoulder dystocia (*n* = 4, 2.6%), macrosomia >4 kg (*n* = 5, 3.2%), breech delivery (*n* = 6, 3.9%), fetal distress (*n* = 8, 5.2%), maternal diabetes (*n* = 2, 1.3%), clavicular fracture (*n* = 2, 1.3%), and forceps delivery (*n* = 4, 2.6%). The comparison with the recent literature suggested that most perinatal OBPP risk factors were within the normal range, although the incidence of shoulder dystocia, forceps and vaginal breech deliveries exceeded the average rates. Conclusion: Perinatal factors related to OBPP did not occur in a higher frequency in patients with Walch type B OA compared to the general population, although some of them were in the high normal range.

## 1. Introduction

Obstetric brachial plexus palsy (OBPP) is a flaccid paresis of the upper extremity with an incidence that ranges from 0.42 to 5.1 per 1000 live births [1,2,3,4,5,6,7]. A common sequela is an internal rotation contracture of the shoulder that frequently leads to developmental disturbances including excessive retroversion of the glenoid [8]. The etiology of OBPP is presumed to be traumatic traction of the brachial plexus nerve roots during childbirth. Neurogenic injury results in muscle and joint imbalance that leads to morphologic changes that progress with time, depending on the severity of the lesion. It has been shown that muscle imbalance between the subscapularis and posterior cuff is associated with posterior subluxation and retroversion of the glenohumeral joint [9].

Glenohumeral osteoarthritis (OA) with a type B retroverted glenoid with erosion is also believe to be initiated by progressive posterior subluxation of the humeral head [10]. The exact cause of such translation has not been elucidated yet, although many anatomical parameters have been identified as potential etiologic factors, including increased premorbid retroversion [11], decreased humeral retrotorsion [12], proximal humeral morphology [13], altered acromial roof morphology, position of the scapula on the thorax, as well as other factors like repetitive dynamic posterior subluxation [14], and muscular disbalance [9]. A primum movens that could explain all previously mentioned type B associations is the presence of a subclinical neurological lesion, such as a OBPP.

The senior authors have stated on the podium, of their anecdotal experiences with type B OA patients reporting a more frequent history of perinatal problems. As such, the senior authors theorize that perinatal issues may be etiologic factors in the development of type B glenoids. At present, however, the senior authors’ beliefs are level 5 evidence and unscientifically supported. Therefore, the purpose of this retrospective observational study was to provide scientific evidence that patients with a Walch type B pattern of osteoarthritis have a higher rate of risk factors for OBPP, compared to reported data from the general population. Our hypothesis was that type B patients would have a substantially higher rate of OBPP risk factors leading to a higher rate of subclinical neurological lesions during childbirth, which would be the etiologic factor for posterior subluxation seen in type B glenoid OA.

## 2. Experimental Section

### 2.1. Study Population and Patient Recruitment

The authors (AL, BE, LF, GSA) retrospectively evaluated a consecutive series of 157 patients (190 shoulders) presenting to their respective institutions between May 2018 and August 2019 with static posterior subluxation of the humeral head, but without any clear sign of OBPP lesions. All patients presented to the orthopedic surgeon for shoulder problems and were prescribed a computed tomography (CT) scan. Inclusion criteria was patients referred to one of these reference centers in the context of Walch type B glenoid (B1 to B3) including static posterior subluxation [15] determined by unilateral or bilateral CT [10]. A bilateral CT scan was systematically performed if the patient symptomatology was suggestive of a shoulder pathology on both sides. Patients were excluded if they were younger than 16 years (*n* = 1) or if they had incomplete documentation (*n* = 2). The final cohort comprised 154 patients (187 shoulders) aged 63 ± 17 years (median, 68; range, 17–89), with a greater proportion of men (68%).

### 2.2. Questionnaire

Patients were asked to complete a questionnaire to determine their age, gender, and a perinatal history related to OBPP, including macrosomia >4 kg, shoulder dystocia, fetal distress/hypoxia (pH < 7.1), maternal diabetes, vaginal breech delivery, clavicular fracture and forceps delivery [16,17] To obtain relevant data, patients were asked to question their biologic parents to ensure completeness.

### 2.3. Literature Review

Since the present study is not comparative, the authors performed a thorough systematic review to assess the mean incidence of the aforementioned perinatal risk factors for OBPP in the general population. The authors used the Preferred Reporting Items for Systematic Reviews and Meta-Analysis (PRISMA) guidelines throughout the literature review and analyses. The literature search was carried out using the PubMed/MEDLINE and Cochrane library databases (date of access: 8 February 2021) using the MeSH terms: “registry”, “nationwide”, “fetal distress”, “birth hypoxia”, “fetal hypoxia”, “macrosomia”, “breech delivery”, “breech presentation”, “fetal presentation”, “forceps”, “delivery”, “shoulder dystocia”, “gestational diabetes”, “clavicle fracture” and “clavicular fracture” in combination with the “AND” or “OR” Boolean operators. The authors also selected relevant studies that were specifically focused on OBPP risk factors, using the MeSH terms: “risk factors”, “brachial plexus birth palsy”, “brachial plexus palsy”, “brachial plexus injur*”, “brachial plexus birth injury”. The inclusion criteria were (1) studies published between January 2016 and January 2021 on (2) peer-reviewed scientific journals. The exclusion criteria were (1) non-english language publications, (2) studies with a different definition of macrosomia (e.g., 4.5 kg threshold), (3) studies in which the incidence of the factors of interest was not available and (4) studies of limited cohort size (<1000 patients).

### 2.4. Ethical Approval

The study protocol was approved by the hospital ethics committee (AMG 12-26), and all patients gave informed consent.

### 2.5. Statistical Analyses 

For baseline characteristics, variables were reported as mean ± standard deviation or proportions, and were compared to recent studies on registries or nationwide databases published in the literature. The authors used R version 3.6.2 (R Foundation for Statistical Computing, Vienna, Austria) to summarize the literature findings, describe the characteristics of the present patient series (descriptive statistics) and create the forest plot illustration. 

## 3. Results

A substantially greater number of patients had unilateral (*n* = 79%) posterior humeral head subluxation with OA versus bilateral (Figure 1). Of the 154 patients, twenty-seven patients (18%) reported one or more perinatal risk factors for OBPP including fetal distress/hypoxia (*n* = 8, 5.2%), breech delivery (*n* = 6, 3.9%), macrosomia > 4 kg (*n* = 5, 3.2%), forceps delivery (*n* = 4, 2.6%), shoulder dystocia (*n* = 4, 2.6%), gestational diabetes (*n* = 2, 1.3%), and clavicular fracture (*n* = 2, 1.3%) (Table 1). 

The literature search yielded 449 articles after duplicates removal (Figure 2). More than 75% of these articles were excluded following title or full-text screening (*n* = 259 and *n* = 81, respectively), leaving a total of 109 studies for further analyses (Appendix A) [16,18,19,20,21,22,23,24,25,26,27,28,29,30,31,32,33,34,35,36,37,38,39,40,41,42,43,44,45,46,47,48,49,50,51,52,53,54,55,56,57,58,59,60,61,62,63,64,65,66,67,68,69,70,71,72,73,74,75,76,77,78,79,80,81,82,83,84,85,86,87,88,89,90,91,92,93,94,95,96,97,98,99,100,101,102,103,104,105,106,107,108,109,110,111,112,113,114,115,116,117,118,119,120]. The comparison with the recent literature suggests that most perinatal OBPP risk factors were comparable to the average rates in the general population (Figure 3, Table 1), except for forceps delivery (2.6% vs. 1.1% ± 0.9%), vaginal breech delivery (3.9% vs. 0.9% ± 1.4%) and shoulder dystocia (2.6% vs. 0.9% ± 0.9%). Although those three factors appear to be more present in our series, they remain however equal of lower than the highest rates reported in the literature (forceps delivery: 2.6%, vaginal breech delivery: 6.0% and shoulder dystocia: 2.8%) [35,75].

## 4. Discussion

Overall, the principal finding of the study was that patients with B glenoid OA tend to have a similar rate of OBPP-related risk factors compared to the general population. The incidence of shoulder dystocia, forceps delivery, and vaginal breech delivery were above the average rates reported in the literature, though not exceeding the highest values. As such, based on the level 4 evidence provided by the results in a small number of patients, the senior authors’ assumptions based on anecdotal level 5 evidence were incorrect.

Delivery can be a traumatic experience for both the mother and the newborn, which could consequently lead to trauma damaging tissues and organs of the newly delivered child [121]. It can supervene as a result of physical pressure, traction on the upper limbs or neck [122]. It encompasses the durable side effects of pregnancy, labor and birth injuries, including the ensuing compensatory and adaptive mechanisms, as well as the development of pathological processes after damage. Clinically evident OBPP remains fortunately rare.

Forces asymmetry created in the glenohumeral joint, as seen in OBPP, affects joint development and lead to glenoid deformity [123]. There is a widespread opinion that undergrowth of the glenohumeral joint in OBPP develops gradually over time, as a consequence of an internal rotation contracture and muscular imbalance [123]. With severe lesions, there might be an increasing posterior displacement of the humeral head with development of a glenoid dysplasia (C glenoid, Figure 3A), whereas in subclinical impairment, a milder deformity (B glenoid, Figure 3B) might develop over decades.

Anatomic (glenoid retroversion, humeral retrotorsion, acromial roof morphology) and muscular (disbalance) factors have all been associated with posterior static subluxation. Interestingly, all the above-mentioned morphologic disturbances are well-known sequelae of OBPP [124]. It may indicate that such changes are related to each other and that they may emerge from one common origin: subclinical OBPP.

It is worth noting that static posterior subluxation of the humeral head was bilateral for 21% of the cases. For these cases, subclinical OBPP is unlikely and condition might be related to other injuries or intense activities such as bench pressing [125].

This study has several limitations including its retrospective and multicentric design making it difficult to gather data, absence of a control group, and the small sample size compared to recent registry or nationwide studies. Further comparative studies with larger cohorts are therefore needed to confirm our findings. Moreover, precise tracking progress during labor, circumstances of the delivery and potential perinatal problems may not be remembered by patients. Nonetheless, birth books are generally kept by patients. We also asked routinely patients to question their parents about delivery conditions. Perinatal problems make parents feeling guilty, leading to psychologically and emotionally damage. Such experience is usually shared with descendants. However, this study has several strengths including the homogeneity of the glenoid morphology analyzed, and its extensive literature review on recent registry and nationwide studies with considerable cohort sizes.

## 5. Conclusions

Perinatal factors related to OBPP did not occur in a higher frequency in patients with Walch type B OA compared to the general population, although some of them were in the high normal range. These findings may indicate that other studies are needed to further investigate the possible association between subclinical neurological lesions during childbirth and static posterior subluxation of the humeral head.

## Figures and Tables

**Figure 1 jcm-10-01196-f001:**
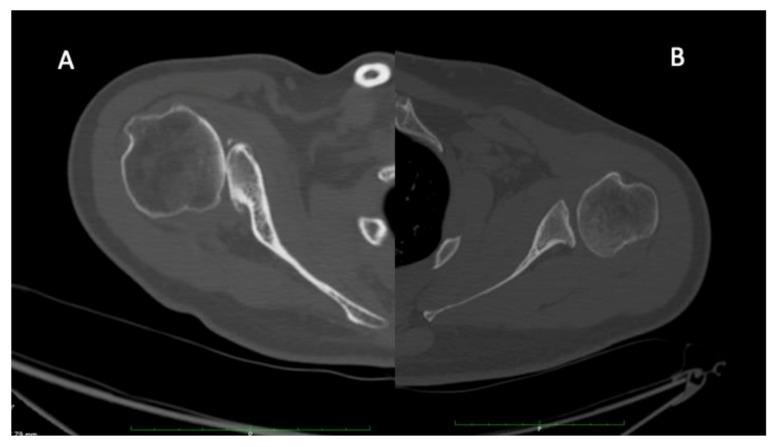
Illustration of C (**A**) and B (**B**) glenoids. C glenoid observed in severe neurological lesion are characterized by a retroversion above 25 degrees, a humeral head subluxation above 80%, a rounded posterior rim, a “lazy J” sign and a hypoplastic neck. On the other hand, B glenoid are characterized by a retroversion above 15 degrees, a humeral head subluxation above 60%, a concave glenoid, osteophytes, a sharp posterior rim and a normal glenoid neck.

**Figure 2 jcm-10-01196-f002:**
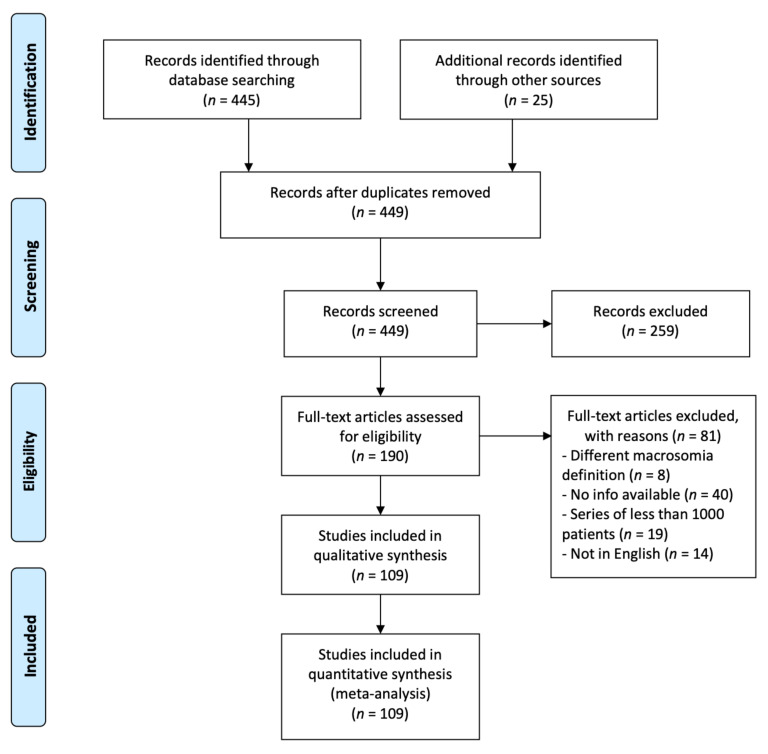
PRISMA flow diagram summarizing the literature search and article screening process.

**Figure 3 jcm-10-01196-f003:**
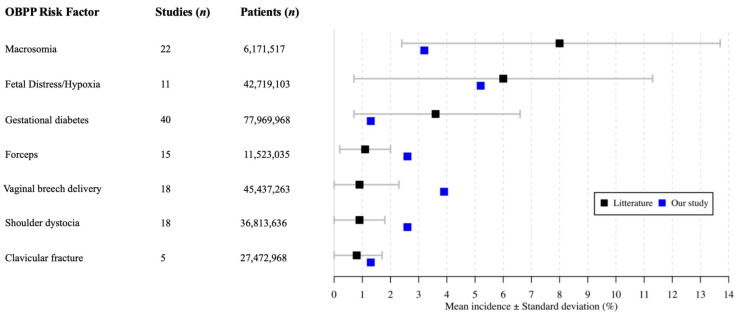
A forest plot comparing the incidence of OBPP-related risk factors between the general population (literature review) and patients with Walch type B shoulder OA (the present series).

**Table 1 jcm-10-01196-t001:** Perinatal risk factors for OBPP.

	Present Study (*n* = 154 Patients) *		Literature Review **				
	*n*	(%)	Mean	±SD	Median	(IQR)	(Min–Max)
Macrosomia > 4 kg	5	(3.2%)	8.0%	±5.7%	7.7%	(3.1%–10.9%)	(0.1%–19.7%)
Fetal distress (hypoxia)	8	(5.2%)	6.0%	±5.3%	5.0%	(1.5%–7.3%)	(0.4%–16.4%)
Gestational diabetes	2	(1.3%)	3.6%	±2.9%	2.9%	(1.4%–4.9%)	(0.2%–15.2%)
Forceps delivery	4	(2.6%)	1.1%	±0.9%	0.9%	(0.3%–1.8%)	(0.1%–2.6%)
Vaginal breech delivery	6	(3.9%)	0.9%	±1.4%	0.4%	(0.2%–0.8%)	(0.1%–6.0%)
Shoulder dystocia	4	(2.6%)	0.9%	±0.9%	0.4%	(0.2%–1.5%)	(0.1%–2.8%)
Clavicular fracture	2	(1.3%)	0.8%	±0.9%	0.2%	(0.2%–1.1%)	(0.1%–2.3%)

OBPP, obstetrical brachial plexus palsy; SD, Standard deviation; IQR, Interquartile Range. *, in patients suffering from B glenoid osteoarthrosis **, in the general population.

## Data Availability

Details regarding where data supporting reported results can be asked at the following e-mail address: hugo.bothorel@latour.ch.

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
