# Peer review of "Could Subtle Obstetrical Brachial Plexus Palsy Be Related to Unilateral B Glenoid Osteoarthritis?"

_jcm, 2021, doi:10.3390/jcm10061196_

Round 1

Reviewer 1 Report

This manuscript has been revised. I have 3 minor comments:
The manuscript is based on the PRISMA guidelines which should be mentioned in the abstract
Figures does not belong in "discussions" and figure 3 should be placed in the "results" section.
Table 2 is rather extensive, and I believe it should be moved to "supplementary materials"  

Author Response

Reviewer 1

Comments

Answers/Corrections

Line

Suggestions for Authors

This manuscript has been revised. I have 3 minor comments:
1- The manuscript is based on the PRISMA guidelines which should be mentioned in the abstract     2- Figures does not belong in "discussions" and figure 3 should be placed in the "results" section.
3- Table 2 is rather extensive, and I believe it should be moved to "supplementary materials".

1- Added: “A literature review was performed following the Preferred Reporting Items for Systematic Review and Meta-Analysis (PRISMA) to estimate the incidence of OBPP risk factors in the general population.”

2- Figures no longer belong in discussions. The Figure 3 was moved to the results section.

3- We completely agree. The Table 2 was moved to supplementary materials since important literature results are already summarized in the Table 1.

1- Lines 26-8

2- Lines 138-45

3- Supplementary Materials

Reviewer 2 Report

The authors thoroughly addressed the suggestions provided in the previous review, improving the quality of their report. Yet, there are some small points that need improvement.

I think it would be advisable to better define the study population: were they patients presenting to the orthopaedic surgeon for shoulder problems and were prescribed a CT scan? Or were they selected among the CT scans performed at the institution?

In the methods section, it is stated that unilateral or bilateral CT scans were performed, but in the results section a percentage of bilateral disease is provided. Could you, please, clarify how the diagnosis was made?

In line 134, referral to the use of R has been made, but the kind of computations made with the program is not explained. Could you, please, enhance this section?

Author Response

Reviewer 2

Comments

Answers/Corrections

Line

Suggestions for Authors

The authors thoroughly addressed the suggestions provided in the previous review, improving the quality of their report. Yet, there are some small points that need improvement. 1- I think it would be advisable to better define the study population: were they patients presenting to the orthopaedic surgeon for shoulder problems and were prescribed a CT scan? Or were they selected among the CT scans performed at the institution? 2- In the methods section, it is stated that unilateral or bilateral CT scans were performed, but in the results section a percentage of bilateral disease is provided. Could you, please, clarify how the diagnosis was made? 3- In line 134, referral to the use of R has been made, but the kind of computations made with the program is not explained. Could you, please, enhance this section? 

Thank you for your comments. Please see our answers below.

1- Specified: “All patients presented to the orthopaedic surgeon for shoulder problems and were prescribed a CT scan.”

2- Specified: “A bilateral CT scan was systematically performed if the patient symptomatology was suggestive of a shoulder pathology on both sides.”

3- Explained: “to summarize the literature findings, describe the characteristics of the present patient series (descriptive statistics) and create the forest plot illustration.”

1-Lines 84-5

2- Lines 88-9

3- Lines 126-9

Reviewer 3 Report

Overview:

The authors have designed a retrospective evaluation of 187 shoulders with static posterior subluxation of the humeral head but without clear signs of OBPP lesions.

Strengths

  • Overall, this study addresses an important and interesting subject
  • Relatively large cohort for this specific problem (type B glenoid)
  • Good comparison to the literature

Although no definitive correlation was found between OBPP and Walch type B  OA the hypothesis make a lot of sense and as concluded other studies are needed to further investigate the possible association between  subclinical neurological lesions during child

Weaknesses

As stated in the limitations paragraph

ALarger cohort is needed for this type of study

 Table 2

Is too long and should be presented in some other fashion

Author Response

Reviewer 3

Comments

Answers/Corrections

Line

Comments and Suggestions for Authors

Overview: The authors have designed a retrospective evaluation of 187 shoulders with static posterior subluxation of the humeral head but without clear signs of OBPP lesions.

/

Strengths·       Overall, this study addresses an important and interesting subject·       Relatively large cohort for this specific problem (type B glenoid)·       Good comparison to the literature Although no definitive correlation was found between OBPP and Walch type B  OA the hypothesis make a lot of sense and as concluded other studies are needed to further investigate the possible association between  subclinical neurological lesions during child 

Thank you

Weaknesses As stated in the limitations paragraph a larger cohort is needed for this type of study 

We agree. Limitation acknowledged.

Table 2Is too long and should be presented in some other fashion 

As suggested by the reviewer 1, the Table 2 was moved to supplementary materials since important literature results are already summarized in the Table 1.

Supplementary materials

This manuscript is a resubmission of an earlier submission. The following is a list of the peer review reports and author responses from that submission.

Round 1

Reviewer 1 Report

This body of this work aims to determine the incidence of OBPP risk factors in patients with glenoid type B osteoarthritis. The assessment and treatment of the glenoid morphologic changes is challenging and requires attention.

The manuscript has an interesting hypothesis but provides little new information on the issue of glenoid dysplasia in osteoarthritic patients.

taken into consideration the methodological restraints.

Introduction:

The Incidence of OBPL is a little higher than reported: 1-5/1000 births Ref : Paediatr Child Health. 2006 Feb;11(2):93-100. Perinatal brachial plexus palsy. Andersen J1, Watt J, Olson J, Van Aerde J.

Line 43: The etiology of OBPP is presumed to be traumatic traction of the brachial plexus nerve roots during childbirth. Neurogenic injury results in muscle and joint imbalance that leads to morphologic changes that progress with age.

It is unclear what the authors here refer to. Waters and Bae (JBJS 2008) suggested that glenohumeral deformity may appear early in life, progress with age and correlates with internal rotation contracture. Glenoid dysplasia in OBPL usually is present as early as a few months after debut but the association between age and morphologic changes is not fully understood.

Line 45: Commonly, an internal rotation contracture occurs due to muscle imbalance between the subscapularis and posterior cuff with resultant posterior subluxation and retroversion of the glenohumeral joint.

The authors should give an explanation about how excessive retroversion is caused by muscle imbalance. Importance of glenoid ossification centers could strengthen this argument.

Methods:

The presentation of data characterises a review or meta-analyses and does not follow traditional guidelines.

A minor concern that I have is the interview part, which is qualitative at best.

Results:

Validity of data could be questioned. Not many patients 60+ years of age remembers if the sustained a clavicular fracture during delivery. Etc.

Results:

R data missing

Discussion:

Is short and precise.

Figures are missing.

The reader can profit from a figure showing glenoid dysplasia in both cases

Author Response

Response to Reviewers

Date: Feb 1st, 2021

Reviewer 1

Comments

Answers/Corrections

Line

Introduction

This body of this work aims to determine the incidence of OBPP risk factors in patients with glenoid type B osteoarthritis. The assessment and treatment of the glenoid morphologic changes is challenging and requires attention. The manuscript has an interesting hypothesis but provides little new information on the issue of glenoid dysplasia in osteoarthritic patients. taken into consideration the methodological restraints.

-> The Incidence of OBPL is a little higher than reported: 1-5/1000 births Ref : Paediatr Child Health. 2006 Feb;11(2):93-100. Perinatal brachial plexus palsy. Andersen J1, Watt J, Olson J, Van Aerde J.

Modified

Line 66

Line 43: The etiology of OBPP is presumed to be traumatic traction of the brachial plexus nerve roots during childbirth. Neurogenic injury results in muscle and joint imbalance that leads to morphologic changes that progress with age. It is unclear what the authors here refer to. Waters and Bae (JBJS 2008) suggested that glenohumeral deformity may appear early in life, progress with age and correlates with internal rotation contracture. Glenoid dysplasia in OBPL usually is present as early as a few months after debut but the association between age and morphologic changes is not fully understood.

We agree with this comment. Severe (clinically obvious lesions) OBPP might be present early as a few months after debut. It may take a longer time in case of subtle lesion.

We change the manuscript as follows:  The etiology of OBPP is presumed to be traumatic traction of the brachial plexus nerve roots during childbirth. Neurogenic injury results in muscle and joint imbalance that leads to morphologic changes that progress with time, depending of the severity of the lesion.

Line 71

Line 45: Commonly, an internal rotation contracture occurs due to muscle imbalance between the subscapularis and posterior cuff with resultant posterior subluxation and retroversion of the glenohumeral joint. The authors should give an explanation about how excessive retroversion is caused by muscle imbalance. Importance of glenoid ossification centers could strengthen this argument. 

The ratio of the infraspinatus and teres minor volume to the subscapularis volume is associated with glenoid morphology, retroversion, and humeral-head subluxation. This suggests that an axial plane force imbalance is associated with

glenoid deformity. (doi :10.2106/JBJS.19.00086) However, the importance of ossification center is not yet clearly defined.

We change for: It has been shown that muscle imbalance between the subscapularis and posterior cuff is associated to posterior subluxation and retroversion of the glenohumeral joint. (doi : 10.2106/JBJS.19.00086) 

Lines 71-2

Methods

The presentation of data characterises a review or meta-analyses and does not follow traditional guidelines. A minor concern that I have is the interview part, which is qualitative at best. 

The authors re-performed a thorough literature review according to the PRISMA method.

Lines 124-144

Results

Validity of data could be questioned. Not many patients 60+ years of age remembers if the sustained a clavicular fracture during delivery. Etc.

We agree with this comment. However, we believe that the collected data are representative for the following reasons. First, the birth book is generally kept by patients and we systematically asked for. Second, we asked routinely patients to questions their parents about delivery conditions. Third, a clavicular fracture during delivery makes parents feeling guilty, leading to psychologically and emotionally damage. Such experience is usually shared with descendants.

We added this information in the limitations of the study

Lines 248-251.

R data missing 

We have to admit that we did not fully understand the question. We are sorry for this and will enjoy answering it if formulated differently. If it refers to the version of R, the information is already available.

Line 153

Discussion

Is short and precise.

Figures are missing.

The reader can profit from a figure showing glenoid dysplasia in both cases

Thank you. The authors added one figure.

The legend of the figure is the following:

Illustration of C (A) and B (B) glenoids. C glenoid observed in severe neurological lesion are characterized by a retroversion above 25 degrees, a humeral head subluxation above 80%, a rounded posterior rim, a “lazy J” sign and a hypoplastic neck. On the other hand, B glenoid are characterized by a retroversion above 15 degrees, a humeral head subluxation above 60%, a concave glenoid, osteophytes, a sharp posterior rim and a normal glenoid neck.

Figure 3 lines 224-231.

Reviewer 2

Comments

Answers/Corrections

Line

The paper explores the relation between subclinical obstetric brachial plexus palsy and developement of type B glenoid osteoarthritis. The underlying hypotesis is that a subclinical neurological damage to the shoulder could cause a muscular imbalance leading to posterior migration of the humeral head and degenerative changes. Unfortunately, risk factors for obstetric brachial plexus palsy and, therefore, for subclinical neurological damage, were not found in higher prevalence in a cohort of patients with a type B glenoid.The paper is generally well written, yet there are a few points that I think need to be clarified. 

 Thank you for your comments. Please see our answers below.

Introduction

The introduction is appropriate, but the aim of the study is expressed only as a hypotesis. I feel that the aim of the study should be plainly stated. 

We added a third paragraph in the introduction

Lines 93-102

Material and methods

In the Patients and Methods section, I suggest improving the description of patients' recruitment and inclusion criteria: no information about the involved centres is provided, the reason why patients presented to the institutions is not provided and the reason for the prescription of the CT scan is not stated. Furthermore, no information is given about the CT scanning protocol: did every patient undergo a bilateral shoulder CT scan? 

We added this information.

Lines 105-112.

Osteoarthritis with a type B glenoid is uncommon in young patients, that are, anyway, included in the study cohort. An explanation for this should be provided. 

Thank you for this interesting remark. Patients with posterior static subluxation are usually young and present after a while B glenoid. This category of patient was included as well. This has been specified if the manuscript.

Line 108-112.

It is stated that data regarding the involved side are collected, but they seem not to be reported in the results section nor elsewhere. 

deleted

Line 118

Accurate results of the literature review are provided in the Results section. Since the data obtained from this review are an essential part of this paper, details on the methods of the literature review should be provided in the methods section. 

We re-performed the literature review according to the PRISMA guidelines

Lines 124-144

Figure 1

Results

The Results section is clear and concise and the tables are well built. 

Thank you

Discussion

Results are extensively discussed in the light of available literature. Limitations of the study are just listed in the text; a deeper analysis would help direction future investigations.

Limitations have been added or better explained. Please refer to the above questions.

Lines 245-51

Some minor spelling errors can be found throughout the text.  

Some spelling errors have been corrected (i.e clavicular line 120).

 Line 120

Tables 1 and 2

Reviewer 3

Comments

Answers/Corrections

Line

The article attempts to explore the cause of static posterior subluxation and B2 Glenoids.  They have done a reasonable job of exploring BBPP as a contribution. While it does not prove that Hypothesis it is of clinical value in that it calls attention again to the idea that muscular imbalance likely causes the subluxation and wear pattern even if the cause of the imbalance is not yet understood. The article includes a good bibliography to have available in the literature for future studies to access. Of course it would have been preferable to have prospective or actual reports of details of the birthing rather than just anecdotal from the patients and their parents  

Thank you

Reviewer 2 Report

The paper explores the relation between subclinical obstetric brachial plexus palsy and developement of type B glenoid osteoarthritis. The underlying hypotesis is that a subclinical neurological damage to the shoulder could cause a muscular imbalance leading to posterior migration of the humeral head and degenerative changes. Unfortunately, risk factors for obstetric brachial plexus palsy and, therefore, for subclinical neurological damage, were not found in higher prevalence in a cohort of patients with a type B glenoid.

The paper is generally well written, yet there are a few points that I think need to be clarified.

The introduction is appropriate, but the aim of the study is expressed only as a hypotesis. I feel that the aim of the study should be plainly stated.

In the Patients and Methods section, I suggest improving the description of patients' recruitment and inclusion criteria: no information about the involved centres is provided, the reason why patients presented to the institutions is not provided and the reason for the prescription of the CT scan is not stated. Furthermore, no information is given about the CT scanning protocol: did every patient undergo a bilateral shoulder CT scan?

Osteoarthritis with a type B glenoid is uncommon in young patients, that are, anyway, included in the study cohort. An explanation for this should be provided.

It is stated that data regarding the involved side are collected, but they seem not to be reported in the results section nor elsewhere. 

Accurate results of the literature review are provided in the Results section. Since the data obtained from this review are an essential part of this paper, details on the methods of the literature review should be provided in the methods section.

The Results section is clear and concise and the tables are well built.

Results are extensively discussed in the light of available literature. Limitations of the study are just listed in the text; a deeper analysis would help direction future investigations.

Some minor spelling errors can be found throughout the text. 

Author Response

(The authors gave the same response as above.)

Reviewer 3 Report

The article attempts to explore the cause of static posterior subluxation and B2 Glenoids.    They have done a reasonable job of exploring BBPP as a contribution.

While  it does not prove that Hypothesis it is of clinical value in that it calls attention again to the idea that muscular imbalance likely causes the subluxation and wear pattern even if the cause of the imbalance is not yet understood. 
The article includes  a good bibliography to have available in the literature for future studies to access.

Of course it would have been preferable to have prospective or actual reports of details of the birthing rather than just anecdotal from the patients and their parents 

Author Response

(The authors gave the same response as above.)
